# Awareness of treatment: A source of bias in subjective grading of ocular complications

**Genis Cardona***◉, **Noelia Esterich**◉

Optics and Optometry Department, Universitat Politècnica de Catalunya, Terrassa, Spain

◉ These authors contributed equally to this work.
* genis.cardona@upc.edu

## Abstract

### Purpose

Bias has been described as one important obstacle in scientific research. The aim of this study was to explore "awareness of treatment" as a possible source of bias in subjective grading of ocular complications.

### Methods

Thirty subjects with similar, basic experience with grading scales participated in the study. The Efron grading scales were used to grade 24 images of three different ocular conditions (eight images each of bulbar hyperaemia, limbal vascularization and corneal staining). Three consecutive, two weeks apart, grading sessions were scheduled, in which the same images were graded, although in the third session images were deceptively labelled as either "treated" or "untreated". Grading results from the first and second sessions were compared to determine grading reliability and discrepancies with the third session informed of grading bias originating from "awareness of treatment".

### Results

Moderate to good test-retest reliability was found for all conditions, with median intraclass correlation values of 0.80 (0.62–0.84) for bulbar hyperaemia, 0.68 (0.65–0.77) for limbal vascularization and 0.68 (0.66–0.74) for corneal staining. Grading values from the first and third sessions evidenced negative and positive systematic errors (bias) for "treated" and "untreated" conditions, respectively. Statistically significant differences were found between the average grading discrepancies of session 1 and session 2 and those of session 1 and session 3 (all p<0.001).

### Conclusions

"Awareness of treatment" may be considered a source of bias of subjective grading of ocular complications, although the actual effect of bias is unlikely to be of clinical significance.

**Data Availability Statement:** Data are available via Figshare (DOI: 10.6084/m9.figshare.8299700).

**Funding:** The authors received no specific funding for this work.

**Competing interests:** The authors have declared that no competing interests exist.

## Introduction

The use of grading scales to assess the presence and severity of ocular complications has increased in recent years. A survey of 237 Australian optometrists revealed that 61% of them employed grading scales in their daily routine [1], with a preference for the artistic Efron Grading Scales for Contact Lens Complications [2]. A second, global survey of 809 eye care practitioners, with a majority of optometrists, disclosed that 84.5% of respondents used grading scales to record ocular findings, with a 51.6% opting for the Efron scales and 48.5% for the Brien Holden Vision Institute scales (formerly known as CCRLU grading scales [3]), which consist of real photographs of ocular conditions [4]. Less recent surveys of UK and Australian ophthalmologists indicated lower rates of use of grading scales for the evaluation of cataract and pterygium, respectively [5,6].

Currently, there are many grading scale options available to the clinician, from those assessing a variety of ocular conditions, both contact lens and non-contact lens related, to scales developed to grade specific disorders, such as lid wiper [7], corneal and conjunctival staining [8,9] or Meibomian gland dysfunction [10], amongst others. In an attempt to avoid the intrinsic limitations of subjective grading, research efforts have been directed to developing objective grading techniques, mainly based on digital image processing [11–14]. However, objective grading techniques are commonly restricted to few ocular conditions, such as bulbar hyperaemia [11,12] or corneal staining [13,14], and are themselves not free of limitations.

Subjective grading of ocular complications is a relatively simple process involving the comparison of real live images, obtained using a slit-lamp, with a set of photographs or artistic drawings representing conditions at various degrees of severity. This process is modulated by the experience, training and knowledge of the observer [15]. In addition to good accuracy and reliability, subjective grading of ocular complications must also be free of bias. Biases have been known to challenge the objectivity of scientific research and, as such, efforts have been devoted to identify, classify and develop strategies to avoid, or at least critically appraise, the impact of biases [16]. An example of such an effort is the extensive, continuously updated catalogue of biases affecting medical evidence compiled by the Centre for Evidence Based Medicine at Oxford University (available at https://catalogofbias.org/). Of interest in subjective grading may be observer bias and misclassification bias. Observer bias refers to the effect of the predispositions or preconceptions of the observer, such as the documented predisposition of examiners to grade to the nearest whole number, even if the scale is divided in 0.1 increments [17,18]. Misclassification bias, on the other hand, occurs when patients are erroneously assigned to a given category, and includes non-differential and differential misclassification bias, the later occurring when the probability of misclassification depends on the actual status of the patient [19,20].

The aim of the present study was to explore differential misclassification bias as an additional source of bias in subjective grading of ocular complications. In particular, a possible source of bias defined as "awareness of treatment". A grading simulation, consisting of three grading sessions, was implemented to explore whether participants graded differently those conditions they were informed were following a successful treatment plan as compared to those left untreated.

## Materials and methods

### Subjects

Optometry students from the Technical University of Catalonia were recruited for this study. Thirty students (17 females) participated in the study, with an age of 21.7±2.1 (mean ± SD)

years, ranging from 21 to 32 years. All subjects were enrolled in the Basic Contact Lenses course, which is programmed in the fifth semester of the 4-year Optics and Optometry degree, and had passed the Ocular Pathology and Pharmacology courses, which are programmed in the fourth semester. All subjects had a basic knowledge of and experience with grading scales for ocular complications. Written informed consent was provided by all participants after the nature of the study was explained to them. In this regards, at the start of the study, subjects were only informed that they would be participating in three grading sessions. It was only following the conclusion of the third grading session that the full purpose of the study was explained, including the partial deceit required for the third grading session. The study was approved by an Institutional Review Board (Facultat d'Òptica I Optometria de Terrassa) (2018-07-27T06).

## Ocular conditions and grading procedure

Three conditions, which may be contact lens or non-contact lens related, were selected for this study: bulbar hyperaemia, limbal vascularization and corneal staining. Photographs were obtained from a database of anterior segment images captured with the same Topcon SL-D7 slit-lamp and DL-4 5-megapixel digital camera (Topcon España S.A., Barcelona, Spain). For image capture, slit-lamp magnification was set at 10X and a circular light beam of 10 mm in diameter was employed to illuminate the ocular surface. To observe and photograph corneal staining, a cobalt blue light filter was used, in combination with a Wratten #12 yellow filter (Kodak, Rochester, NY, US) positioned in front of the observation system of the slit-lamp. Twenty-four images (8 of each condition) were selected, aiming at a wide spectrum of disease severity, thus also including healthy eyes. At the time of the study, images did not include any type of identification linking them to the corresponding original patients.

The Efron Grading Scales for Contact Lens Complications were used. These scales, which are described in detail in the literature [2,21], consist of 16 sets of artistic drawings that cover key anterior ocular complications of contact lens wear, illustrated in five stages of increasing severity from zero to four. The sets depicting bulbar hyperaemia, limbal vascularization and corneal staining were selected.

Subjects used copies of a printed vertical visual analogue scale (VVAS) to grade each of the conditions [22]. This scale consisted of a 100-millimetre vertical line with five markings on its length to designate grades 0 to 4 (that is, at 0, 25, 50, 75 and 100 mm, respectively). Next to each of these markings, the corresponding drawing from the Efron scale was presented to offer visual clues to assist in the grading process. Ocular images were displayed on a 24 inch, 16:9 liquid crystal display (TFT-LCD) set to a resolution of 1920 per 1080 pixels, 32-bit colour configuration, contrast ratio 700:1 and 75 Hz refresh rate. Room illumination conditions were constant throughout the grading sessions. Participants observed the images on the computer screen at a distance of approximately 50 centimetres.

After briefly explaining the grading process, each subject was asked to grade the 24 images. Participants had 30 seconds to grade each condition. Grades were assigned by marking the desired location on the VVAS. Grading scores were obtained by measuring the height of each mark on the VVAS. All measurements were conducted by a research assistant not aware of the purpose of the study.

Three grading sessions were scheduled, with a two-week interval between consecutive sessions. At each grading session, the same 24 images were graded, albeit in different, randomly generated, inter-session and inter-subject sequences of presentation. At the third and last grading session, subjects were deceptively informed that they would be grading the same patients as in the previous two sessions, but that some of them had been following a successful

treatment plant, whereas the conditions of others were left untreated. Accordingly, each image was accompanied by a non-intrusive label indicating either "treated" or "untreated". For each subject and image, "treated" and "untreated" labels were assigned and presented randomly, with 12 images labelled as "treated" and the other 12 images as "untreated". To avoid conscious bias, the relevance of the label on the images was not stressed to the participants.

## Data analysis

Statistical analysis of the data was performed with the IBM SPSS Statistics software 25.0 (IBM Corp., NY, US) for Windows. All data were examined for normality using the Kolmogorov-Smirnov test, which revealed normal distributions for most of the variables. A *p-value* of 0.05 or less was considered to denote statistical significance throughout the study.

Intraclass correlation coefficients (ICC) (two-way mixed effects, absolute agreement, multiple examiners/measurements model) of session 1 and 2 (test-retest) were calculated to determine grading reliability for each image and the corresponding median and range (minimum-maximum) of values for each ocular condition (comprised of 8 images) was determined. In addition, a Bland-Altman analysis was conducted by pooling test-retest data of the eight images of each type of condition. The mean difference (bias) and Limits of Agreement (LOA, defined as the mean difference ± 1.96 standard deviations of the mean difference) were determined, as well as the approximate 95% confidence limits for the LOAs, given a sufficiently large sample size (n = 240) [23,24].

Finally, to explore grading bias resulting from "awareness of treatment", a paired Student's t-test was used to compare the mean grading differences of session 1 and 2 (test-retest) with those of session 1 and 3 (either "treated" or "untreated" status) for each ocular condition. Bland-Altman analysis was conducted to explore and display any systematic bias.

## Results

### Grading reliability

Average values and 95% confidence intervals of test-retest grading differences were 0.2 (-0.4 to 0.8) for bulbar hyperaemia, 0.0 (-0.7 to 0.7) for limbal vascularization and -0.1 (-0.6 to 0.4) for corneal staining. Median (minimum-maximum) ICC test-retest (session 1 and 2) values were 0.80 (0.62–0.84) for bulbar hyperaemia, 0.68 (0.65–0.77) for limbal vascularization and 0.68 (0.66–0.74) for corneal staining. Overall, higher ICC values corresponded to images displaying conditions located near the top or bottom thresholds of the grading scales. **Fig 1A, 1B and 1C** display the Bland-Altman analysis for each condition. Upper and lower LOA were 9.0 (95% CI 8.0 to 9.0) and -8.7 (95% CI -7.7 to -9.7) for bulbar hyperaemia, 10.3 (95% CI 9.2 to 11.5) and -10.4 (95% CI -9.2 to -11.5) for limbal vascularization and 8.9 (95% CI 7.9 to 9.9) and -9.2 (95% CI -8.2 to -10.2) for corneal staining. No noticeable systematic error was observed in grading of any of the three conditions under study. As with the ICC analysis, the Bland-Altman plots show less dispersion of the data towards both ends of the grading spectrum.

### Awareness of treatment

Average values and 95% confidence intervals for grading differences between session 1 and session 3, when considering the "treated" conditions were -6.4 (-7.2 to -5.6) for bulbar hyperaemia, -4.5 (-5.1 to -3.9) for limbal vascularization and -3.6 (-4.7 to -2.4) for corneal staining. In contrast, average grading differences for conditions left "untreated" were 5.3 (4.4 to 6.2) for bulbar hyperaemia, 4.4 (3.7 to 5.1) for limbal vascularization and 3.8 (2.9 to 4.7) for corneal staining. Statistically significant outcomes were found between the average grading differences

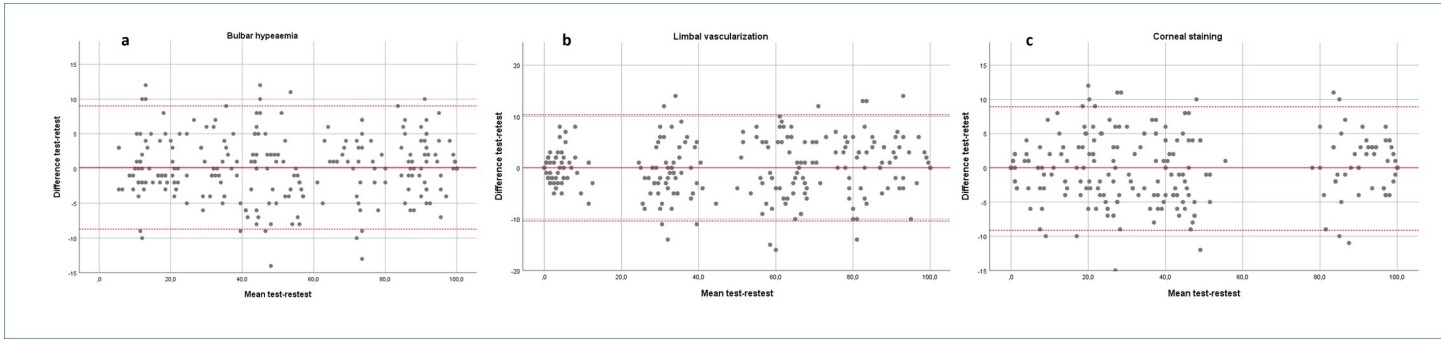

**Fig 1. Bland-Altman analysis of the test-retest grading sessions. 1a**. Bulbar hyperaemia. **1b**. Limbal vascularization. **1c**. Corneal staining. Mean test-retest difference (solid line) and lower and upper limits of agreement are shown (discontinuous lines).

of session 1 and session 2 and those of session 1 and session 3 in bulbar hyperaemia "treated" (t = 16.336; p<0.001) and "untreated" (t = -12.620; p<0.001), limbal vascularization "treated" (t = 10.508; p<0.001) and "untreated" (t = -9.926; p<0.001) and corneal staining "treated" (t = 5.674; p<0.001) and "untreated" (t = -9.759; p<0.001). In all instances, in session 3, "treated" conditions were allocated smaller grading values and "untreated" conditions larger grading values than in session 1. The Bland-Altman analysis for each condition labelled as either "treated" or "untreated" is displayed in **Fig 2**. **Fig 2A** and **2B** correspond to bulbar hyperaemia, with upper and lower LOA values of 6.3 (95% CI 4.3 to 8.3) and -19.1 (95% CI –17.1 to -21.1) for "treated" conditions and of 14.8 (95% CI 13.3 to 16.3) and -4.2 (95% CI -2.7 to -5.7) for "untreated" conditions. Similarly, **Fig 2C** and **2D** display limbal vascularization, with upper and lower LOA values of 3.3 (95% CI 2.0 to 4.5) and -12.3 (95% CI -11.1 to -13.6) for "treated" conditions and of 16.3 (95% CI 14.5 to 18.2) and -7.3 (95% CI -5.4 to -9.2) for "untreated" conditions. Finally, **Fig 2E** and **2F** display the corresponding analysis for corneal staining, with LOA values of 7.5 (95% CI 5.7 to 9.2) and -14.6 (95% CI -12.8 to -16.3) for "treated" conditions and of 13.7 (95% CI 12.2 to 15.3) and -6.0 (95% CI -4.5 to -7.6) for "untreated" conditions. Negative and positive systematic errors were evidenced for "treated" and "untreated" conditions, respectively.

## Discussion

The type of bias described in this study may be included in the differential disease misclassification category, in that the disease is misclassified according to the actual disease status: Examiners are told on the third session that the condition is either "treated" or "untreated", and this leads to a misclassification of the status, in this case the grade they assign [19,20]. Therefore, the initial hypothesis of the present study was that examiners would tend to award higher grades to conditions known to be left "untreated" than to "treated" conditions. In all sessions, examiners were instructed to grade 24 images (eight of each ocular condition under study) and were allowed 30 seconds per image. Efron and McCubbin noted better grading precision when observers were allowed 2 seconds to grade each image, but no further improvement was evidenced when grading time was extended to 60 seconds [25]. Therefore, a grading time of 30 seconds was considered sufficient.

Test-retest reliability, as explored with the ICC and Bland-Altman analysis, revealed a moderate (ICC values between 0.5 and 0.75) to good reliability (ICC between 0.76 and 0.9) for all conditions [26,27], albeit results were inferior to those reported by other authors for bulbar hyperaemia [28,29] and corneal staining [29] grading. Differences in grading procedure, type

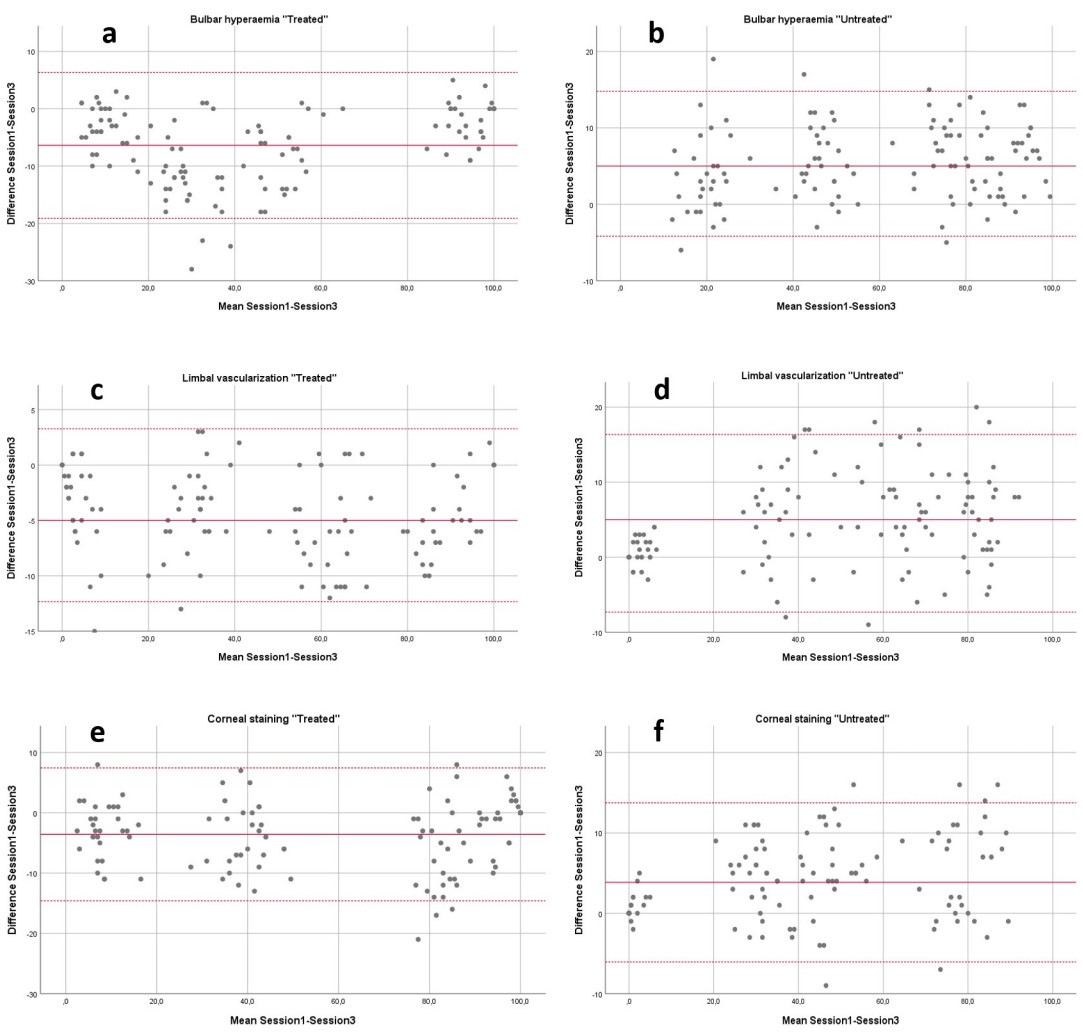

**Fig 2. Bland-Altman analysis of the session1-session3 grading sessions. 2a.** Bulbar hyperaemia "treated". **2b.** Bulbar hyperaemia "untreated". **2c.** Limbal vascularization "treated". **2d.** Limbal vascularization "untreated". **2e.** Corneal staining "treated". **2f.** Corneal staining "untreated". Mean test-retest difference (solid line) and lower and upper limits of agreement are shown (discontinuous lines).

of reference grading scales and knowledge, training and experience of examiners may account for these discrepancies.

Within the same condition, grading reliability was independent of the severity of the condition, with similar individual ICC values and dispersion of data in the Bland-Altman plots. These results are in agreement with those reported by Efron and co-workers [15]. Interestingly, however, in general, higher ICC values were awarded to images displaying conditions located near the top or bottom thresholds of the grading scales. As noted by Bailey et al [18], scales with intrinsic upper and lower limits may lead to reduced data dispersion if conditions are near those limits. This phenomenon was particularly manifest for an image of limbal vascularization depicting a healthy eye (values of, or near 0) and for an image of severe corneal staining, which most examiners graded as 100 in the severity scale.

Average values for test-retest grading differences (0.2 for bulbar hyperaemia, 0.0 for limbal vascularization and -0.1 for corneal staining) did not evidence any bias between session 1 and

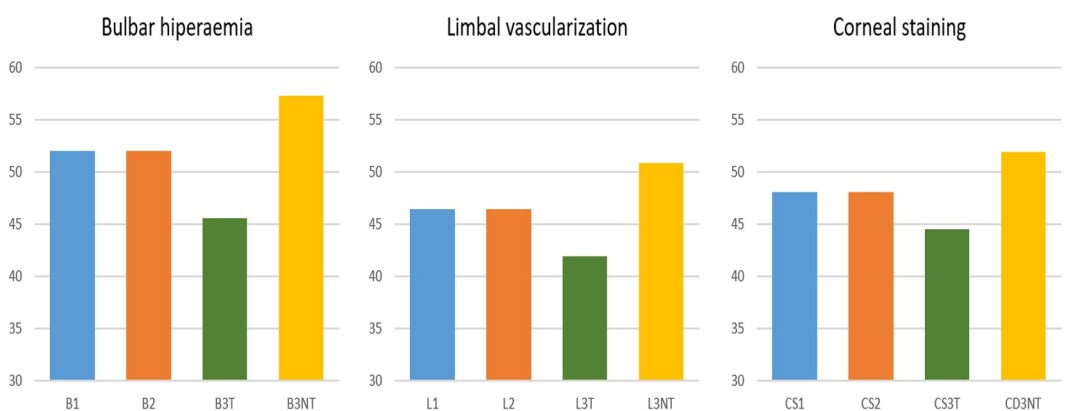

**Fig 3. Mean grading scores of the three sessions (session 3 shows scores for "treated" [T] and "untreated" [NT] images) for each condition (Bulbar hyperaemia: B; Limbal vascularization: L; Corneal Staining: C).**

session 2. In contrast, when comparing session 1 and session 3, a distinctive bias was found in which "treated" conditions were systematically awarded lower grades (-6.4 for bulbar hyperaemia, -4.5 limbal vascularization and -3.6 for corneal staining) and "untreated" conditions higher grades (5.3 for bulbar hyperaemia, 4.4 for limbal vascularization and 3.8 for corneal staining) than the same images at session 1 (**Fig 3**).

It must be acknowledged that, even if presenting statistical significance, grading bias resulting from "awareness of treatment" may not be considered as clinically significant. Indeed, the highest bias (-6.4 for "treated" bulbar hyperaemia) represents a change of -6.4%, with a change as small as -3.6% for "treated" corneal staining. Clinical significance may be defined as the smallest difference between two measures that would compel the clinician to modify his or her decision concerning the management of the patient. Further research is required to determine the threshold of clinical significance associated with changes in the severity of the various ocular conditions in which subjective grading is commonly implemented. It is interesting to note, however, that this study used a modified version of the Efron scale, presented as a continuous vertical line ranging from 0 mm (healthy eye) to 100 mm (highest possible grade), in contrast to the typical 4 or 5-steps scales commonly employed in clinical practice. Previous researchers have observed that continuous scales are associated with higher grading precision than discrete scales [18,29]. It may be relevant to explore the actual effect of "awareness of treatment" when using a discrete scale in which a 1-step difference may represent a 20 or 25% change in grading.

The current study was not devoid of limitations. Firstly, only three typical ocular conditions were assessed. It would be interesting to explore grading bias with an assortment of images depicting conditions offering various degrees of grading challenge to the examiners. Secondly, experience, knowledge and training of the current study sample of students was limited and relatively homogeneous, and not necessarily representative of the whole population of ocular health providers. Indeed, assessing bias in a group of students with inferior skills may increase the effect of the bias (they may be more sensitive to the suggestion of "treated" vs "untreated"), which may artificially inflate the results. Finally, grading was conducted under strictly controlled conditions (time, type of grading scale, image presentation and parameters, etc.) which may not reflect the challenge encountered in grading real-life conditions in a clinical setting. Therefore, findings of this study must be interpreted with caution in views of its limited ecological validity.

In conclusion, many sources of bias have been reported to influence grading precision and reliability. The present findings revealed a statistically significant bias, referred to as "awareness of treatment", in which examiners with moderately reliable grading skills tended to award higher grades to "untreated" conditions and lower grades to "treated" conditions. As the study was designed, with very homogeneous characteristics in sample and grading conditions, the clinical significance of this source of bias could not be considered as manifestly superior to the normal range of variation found when making successive judgments of the same image.

## Author Contributions

**Conceptualization:** Genis Cardona, Noelia Esterich.

**Data curation:** Noelia Esterich.

**Formal analysis:** Genis Cardona.

**Investigation:** Genis Cardona, Noelia Esterich.

**Methodology:** Genis Cardona, Noelia Esterich.

**Validation:** Genis Cardona, Noelia Esterich.

**Writing – original draft:** Genis Cardona, Noelia Esterich.

**Writing – review & editing:** Genis Cardona, Noelia Esterich.

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
