## [Decision Letter · Decision Letter 0]

6 Aug 2019

PONE-D-19-17494

Awareness of disease progression: a source of bias in subjective grading of ocular complications

PLOS ONE

Dear Dr Cardona,

Thank you for submitting your manuscript to PLOS ONE. After careful consideration, we feel that it has merit but does not fully meet PLOS ONE’s publication criteria as it currently stands. Therefore, we invite you to submit a revised version of the manuscript that addresses the points raised during the review process.

Please address all reviewers comments below, including the following issues:

1. The background of the study is not clearly described in the Introduction. How does 'bias' differ from 'variability in grading'? Bias needs to be clearly defined and its sources detailed in the Introduction. There is a recent article on philosophical bias which the authors could also refer to in their deliberations on this area (Andersen et al. eLife 2019;8:e44929. DOI: https://doi.org/10.7554/eLife.44929). This article also notes the Catalogue of Biases (https://catalogofbias.org/about/) which is useful in the context of the current study.

2. The concept of bias is not novel, but rather more appreciated within the clinical and scientific community (see point 1. above).  

3. Please revise the title. The participants are not given details on disease progression, but rather images with descriptors of 'treated' and 'not treated'. The study is thus addressing the impact of awareness of treatment, .

4. Please confirm that consent from the participants was given, and also provide the IRB number in the text.

5. Please see Reviewer 2 comments regarding the statistics used in this study. These comments must be addressed. 

6. All participants were students within an Optometry program, and as noted in the Discussion, this limits the real-world relevance of the study. This issue should be emphasised in the Discussion.

7. There are sections in the Discussion that do not directly relate to the current study (see Reviewer 2). Please review these and remove irrelevant sections.

We would appreciate receiving your revised manuscript by Sep 20 2019 11:59PM. To enhance the reproducibility of your results, we recommend that if applicable you deposit your laboratory protocols in protocols.io, where a protocol can be assigned its own identifier (DOI) such that it can be cited independently in the future. For instructions see: http://journals.plos.org/plosone/s/submission-guidelines#loc-laboratory-protocols

We look forward to receiving your revised manuscript.

Kind regards,

Michele Madigan

Academic Editor

PLOS ONE

Journal Requirements:

2. We noticed you have some minor occurrence of overlapping text with the following previous publication, which needs to be addressed: Cardona, Genís, and Carme Serés. "Grading contact lens complications: the effect of knowledge on grading accuracy." Current eye research 34.12 (2009): 1074-1081. In your revision ensure you cite all your sources (including your own works), and quote or rephrase any duplicated text outside the methods section. Further consideration is dependent on these concerns being addressed.

3. Please provide additional details regarding participant consent. In the ethics statement in the Methods and online submission information, please ensure that you have specified (1) whether consent was informed and (2) what type you obtained (for instance, written or verbal, and if verbal, how it was documented and witnessed). If the need for consent was waived by the ethics committee, please include this information

4. We note that Figure 1  in your submission contains copyrighted images. All PLOS content is published under the Creative Commons Attribution License (CC BY 4.0), which means that the manuscript, images, and Supporting Information files will be freely available online, and any third party is permitted to access, download, copy, distribute, and use these materials in any way, even commercially, with proper attribution. For more information, see our copyright guidelines: http://journals.plos.org/plosone/s/licenses-and-copyright.

Reviewers' comments:

Reviewer's Responses to Questions

**Comments to the Author**

1. Is the manuscript technically sound, and do the data support the conclusions?

Reviewer #1: Partly

Reviewer #2: No

2. Has the statistical analysis been performed appropriately and rigorously? 

Reviewer #1: Yes

Reviewer #2: No

3. Have the authors made all data underlying the findings in their manuscript fully available?

Reviewer #1: Yes

Reviewer #2: No

4. Is the manuscript presented in an intelligible fashion and written in standard English?

Reviewer #1: Yes

Reviewer #2: Yes

5. Review Comments to the Author

Reviewer #1: General Comments

In this manuscript, the authors report on the impact of “awareness of disease progression” as a source of potential bias in the grading of ocular images. The manuscript is generally well written, with some minor comments located below. There are a few scientific questions which arise from the study design which need to be addressed by the authors for this manuscript’s data to be interpreted appropriately.

1) There is the possibility of an order effect, as the labelled images were only presented on the final session. The lack of an appropriate control group which did not have labelled images in the third session make reaching conclusive conclusions regarding the observed change difficult with this as a confound.

2) The modification of the grading scale from an Ordinal/Discreet scale to one that is a more interval based one using a vertical visual analogue scale. This is different that what the scale would be used for clinically, so discussion on the impact of this modification and thus the output data which was evaluated (on a 100 point scale rather than a 0-4 scale) needs to be discussed

3) The use of deception if used typically requires a debrief of subjects after the experiment has concluded, but no details regarding if subjects were debriefed after the study was concluded has been discussed or detailed

Specific Comments

Ethics Statement – Incomplete (no ethics number and type of consent has been detailed)

Introduction

- Line 47 – change “optometrist” to “optometrists”

- Line 60 – Remove “Beside” at the beginning of this sentence

Methods

- How do the authors propose to deal with the confound of the order effect? The changes being seen in the grading may be due to the labelling but may also be due to the learning due to the third test scenario. This could have been alleviated with the use of a control group which did not see these labels, or by randomizing the session where the labels were presented

- Deception: How were subjects debriefed as to the nature and purpose of the deceit?

- What is the impact of using the vertical analogue scale in conjunction with the Efron scale? The Efron scale is an ordinal scale, but the vertical analogue scale converts this to an interval scale which defines the differences between the grades to a certain value (a distance on the scale in this instance). As such, the grading is not exactly the same as grading using simply the Efron scale would be, which only has 5 discreet choices which can be made.

Reviewer #2: Comments are given in the order they occur in the manuscript:

60-62: Citations required.

65: Subjective grading of ocular complications is a complex process involving the comparison of real live images, provided by the slit-lamp….

i. Isn't grading a relatively simple process for the observer? Although the neurological processing involved is undoubtedly complex, it occurs in the background without the observer’s awareness. The whole point of grading systems is that they are simple for clinicians to use, compared to the alternatives.

ii. ....comparison of real-life images, obtained using a slit lamp……

67-72: This is a very complex, composite sentence that is difficult to understand. Can it be simplified?

79-81: What is the difference between having an advanced knowledge of contact lens complications (high intensity) and having been additionally trained in ocular pathology (specificity)? Seems that these would be the same thing.

82: ….the Brien Holden……

92: “The aim of the present study……explore additional sources of bias….”. The Introduction is unsatisfactory because, while it discusses some aspects of using grading scales (intra- and inter- user variability, tendency to use increments etc.), it does little to explain how the outcomes might be subject to bias. Thus, the suggestion here that the aim of this study is “exploring additional sources of bias” is unconvincing. Variability and bias are different phenomena and in the context of this manuscript, the Introduction should be focused on the latter rather than the former.

92: Is the term "awareness of disease progression" the most appropriate to describe what is being assessed here? What the graders are being told is whether that eye has been treated or not. They do not know anything about how the disease is progressing, just that someone has done something. Consequently the title of the paper may be misleading and should be changed.

102-109: Please verify that informed consent was obtained from participants.

Did you inform the participants that they should attempt to carry out the grading according to the scale provided and irrespective of the label on the images? This is an important factor as conscious bias may be more pronounced than unconscious bias.

In general, the statistical analysis seems overly complicated to achieve the stated aims. Bland & Altman (Bland and Altman 1999) have given clear guidelines on how to investigate these issues and as these methods are being used (though without appropriate citations, please add), it isn’t obvious why additional and more convoluted analyses are necessary. For any given condition (or all conditions combined) all that is needed to test the hypothesis that treatment knowledge affects grading scores is:

i. First, show that the first two repeats (R1 and R2) are unbiased, i.e. the mean difference (R1-R2), for all observers, is not significantly different from zero.

ii. Second, show that the third repeat (R3) is significantly biased, by demonstrating that the mean differences (R1-R3) and/or (R2-R3) are significantly different from zero.

Both i) and ii) above can be achieved by observing the 95% confidence interval for the mean differences and inspection of the scatter of points in the plots may add additional insight. It isn’t clear what extra value is offered by ANOVA, correlation, ICCs etc., so these procedures need to be clearly justified, if they are to be included.

191: Why is the precision quoted for limbal vascularization different from the other features (integer vs one decimal place)?

191-194: Can you explain what the difference is between “test-retest differences” and test-retest discrepancy”? These sound like the same thing, but as the SDs being quoted are different, presumably they are not? Is the ”average value of the test-retest differences” not the same as the “average value for the test-retest discrepancy distribution”? Some elaboration and clarification is required.

246-267: This section appears to have nothing to do with the study data or outcomes. Please consider deleting it to shorten the text.

268-289: Likewise, this section repeats information already given elsewhere in the manuscript and can be deleted.

344-351: Although this is presented as the Conclusion, most of the text does not rely on or “conclude” from data presented in the manuscript. For example, the text between lines 344 and 348 makes a series of unrelated statements about clinical practice. Please revise to ensure conclusions are relevant and meaningful.

The use of the word “novel’ at lines 94 and 348 is unreasonable and should be deleted. Types of bias have always existed, it is only our awareness of them that is new.

Check Ref 11: Spelling of Downie.

Bland, J Martin, and Douglas G Altman. 1999. 'Measuring agreement in method comparison studies', Statistical Methods in Medical Research, 8: 135-60.

6. PLOS authors have the option to publish the peer review history of their article (what does this mean?). If published, this will include your full peer review and any attached files.

Reviewer #1: No

Reviewer #2: No

---

## [Author Response · Author response to Decision Letter 0]

19 Sep 2019

Manuscript No.: PONE-D-19-17494

Title: Awareness of disease progression: a source of bias in subjective grading of ocular complications

PLOS ONE

All changes are marked in red in the manuscript. We would like to use this opportunity to express our gratitude to the reviewers for their comments and suggestions, with which we believe our revised manuscript has been improved. 

Academic Editor

Please address all reviewers’ comments below, including the following issues:

1. The background of the study is not clearly described in the Introduction. How does 'bias' differ from 'variability in grading'? Bias needs to be clearly defined and its sources detailed in the Introduction. There is a recent article on philosophical bias which the authors could also refer to in their deliberations on this area (Andersen et al. eLife 2019;8:e44929. DOI: https://doi.org/10.7554/eLife.44929). This article also notes the Catalogue of Biases (https://catalogofbias.org/about/) which is useful in the context of the current study.

ANSWER: Thank you for this comment and for providing us with very relevant sources of information to better address the subject of bias. We have edited our introduction extensively to describe with more detail and thoroughness the background of the study.

2. The concept of bias is not novel, but rather more appreciated within the clinical and scientific community (see point 1. above). 

ANSWER: Indeed, we have attempted to provide a better description of relevant sources of bias. In addition, we have removed the term “novel” from the manuscript.

3. Please revise the title. The participants are not given details on disease progression, but rather images with descriptors of 'treated' and 'not treated'. The study is thus addressing the impact of awareness of treatment.

ANSWER: We have changed the title to: “Awareness of treatment: a source of bias in subjective grading of ocular complications”.

4. Please confirm that consent from the participants was given, and also provide the IRB number in the text.

ANSWER: Consent was obtained from all participants of the study. The IRB details are provided in the text.

5. Please see Reviewer 2 comments regarding the statistics used in this study. These comments must be addressed. 

ANSWER: We have revised our statistical analysis and removed those parts that were redundant in accordance to the suggestions of Reviewer 2.

6. All participants were students within an Optometry program, and as noted in the Discussion, this limits the real-world relevance of the study. This issue should be emphasised in the Discussion.

ANSWER: Although this limitation of our study was already stated in the original manuscript, we have stressed the relevance of this issue.

7. There are sections in the Discussion that do not directly relate to the current study (see Reviewer 2). Please review these and remove irrelevant sections.

ANSWER: As suggested by Reviewer 2, we have removed the irrelevant sections from the discussion of the manuscript. 

We noticed you have some minor occurrence of overlapping text with the following previous publication, which needs to be addressed: Cardona, Genís, and Carme Serés. "Grading contact lens complications: the effect of knowledge on grading accuracy." Current eye research 34.12 (2009): 1074-1081. In your revision ensure you cite all your sources (including your own works), and quote or rephrase any duplicated text outside the methods section. Further consideration is dependent on these concerns being addressed.

ANSWER: We have reviewed the text to identify and address all occurrences of overlapping with the previous publication. In addition, we have provided additional references to the text as appropriate.

Please provide additional details regarding participant consent. In the ethics statement in the Methods and online submission information, please ensure that you have specified (1) whether consent was informed and (2) what type you obtained (for instance, written or verbal, and if verbal, how it was documented and witnessed). If the need for consent was waived by the ethics committee, please include this information

ANSWER: We have provided these details in the manuscript and in the online submission information. In the revised manuscript, we have added the following: “Written informed consent was provided by all participants after the nature of the study was explained to them. In this regards, at the start of the study, subjects were only informed that they would be participating in three grading sessions. It was only following the conclusion of the third grading session that the full purpose of the study was explained, including the partial deceit required for the third grading session.”

We note that Figure 1 in your submission contains copyrighted images. All PLOS content is published under the Creative Commons Attribution License (CC BY 4.0), which means that the manuscript, images, and Supporting Information files will be freely available online, and any third party is permitted to access, download, copy, distribute, and use these materials in any way, even commercially, with proper attribution. For more information, see our copyright guidelines: http://journals.plos.org/plosone/s/licenses-and-copyright.

We require you to either (1) present written permission from the copyright holder to publish these figures specifically under the CC BY 4.0 license, or (2) remove the figures from your submission.

ANSWER: We contacted the publisher (Elsevier) and asked permission to publish Figure 1 under the CC BY 4.0 licence. As the book in which the images appear is published on http://www.sciencedirect.com/, we were automatically instructed the following: 

• Locate the publication containing your desired content on http://www.sciencedirect.com/science/jrnlallbooks

• Click on the article/chapter name to access the abstract

• Below the author details, click “Get Rights and Content” 

• The Rightslink request page will then be launched (please disable your pop-up blocker)

• Select the way you would like to reuse the content

• Create a Rightslink account if you haven’t done so already

• Accept the terms and conditions and you’re done

Whereupon, the corresponding fee to use the image was determined. After careful consideration, we decided to remove Figure 1 from the submission. We have edited the text accordingly.

Reviewer #1

General Comments

1) There is the possibility of an order effect, as the labelled images were only presented on the final session. The lack of an appropriate control group which did not have labelled images in the third session make reaching conclusive conclusions regarding the observed change difficult with this as a confound.

ANSWER: The concern of the reviewer is reasonable and adding a control group to the study would have been interesting. Unfortunately, this is no longer possible. However, we believe that, by conducting three grading sessions we partly covered the possible handicap of lacking a control group. Indeed, our findings revealed a moderate to good grading reliability of our participants, when comparing the first and second grading sessions, with no evidence of bias towards higher or lower grades. It was only on the third grading session that a noticeable bias was observed, in which “treated” conditions were consistently awarded lower grades and “untreated” conditions consistently received higher grades than in previous grading sessions. Although we acknowledge that other confounding factors may be at play, we believe it is reasonable to accept the hypothesis that the observed, and consistent, positive or negative bias, originated in the actual “awareness of treatment”. Otherwise, we would have difficulty in interpreting how the order of grading sessions could lead to such consistent findings.

2) The modification of the grading scale from an Ordinal/Discreet scale to one that is a more interval based one using a vertical visual analogue scale. This is different that what the scale would be used for clinically, so discussion on the impact of this modification and thus the output data which was evaluated (on a 100 point scale rather than a 0-4 scale) needs to be discussed

ANSWER: Although this issue was already briefly addressed in the original manuscript, we have elaborated on the discussion of the possible differences between a continuous 100-point scale and a 0-4 discrete scale. The new paragraph reads as follows: “It is interesting to note, however, that this study used a modified version of the Efron scale, presented as a continuous vertical line ranging from 0 mm (healthy eye) to 100 mm (highest possible grade), in contrast to the typical 4 or 5-steps scales commonly employed in clinical practice. Previous researchers have observed that continuous scales are associated with higher grading precision than discrete scales [18,27]. It may be relevant to explore the actual effect of “awareness of treatment” when using a discrete scale in which a 1-step difference may represent a 20 or 25% change in grading.” 

3) The use of deception if used typically requires a debrief of subjects after the experiment has concluded, but no details regarding if subjects were debriefed after the study was concluded has been discussed or detailed

ANSWER: The reviewer is correct. We have added the following to the Subjects section of the revised manuscript: “Written informed consent was provided by all participants after the nature of the study was explained to them. In this regards, at the start of the study, subjects were only informed that they would be participating in three grading sessions. It was only following the conclusion of the third grading session that the full purpose of the study was explained, including the partial deceit required for the third grading session.” 

Specific Comments

Ethics Statement – Incomplete (no ethics number and type of consent has been detailed)

ANSWER: This information is provided in the revised manuscript. 

Introduction

- Line 47 – change “optometrist” to “optometrists”

- Line 60 – Remove “Beside” at the beginning of this sentence

ANSWER: Thank you. We have edited the text. 

Methods

- How do the authors propose to deal with the confound of the order effect? The changes being seen in the grading may be due to the labelling but may also be due to the learning due to the third test scenario. This could have been alleviated with the use of a control group which did not see these labels, or by randomizing the session where the labels were presented

ANSWER: Please refer to the response provided in the general comments. 

- Deception: How were subjects debriefed as to the nature and purpose of the deceit?

ANSWER: Please refer to the response provided in the general comments.

- What is the impact of using the vertical analogue scale in conjunction with the Efron scale? The Efron scale is an ordinal scale, but the vertical analogue scale converts this to an interval scale which defines the differences between the grades to a certain value (a distance on the scale in this instance). As such, the grading is not exactly the same as grading using simply the Efron scale would be, which only has 5 discreet choices which can be made.

ANSWER: Please refer to the response provided in the general comments.

Reviewer #2

Comments are given in the order they occur in the manuscript

60-62: Citations required.

ANSWER: We have added citations to this sentence.

65: Subjective grading of ocular complications is a complex process involving the comparison of real live images, provided by the slit-lamp….

i. Isn't grading a relatively simple process for the observer? Although the neurological processing involved is undoubtedly complex, it occurs in the background without the observer’s awareness. The whole point of grading systems is that they are simple for clinicians to use, compared to the alternatives.

ii. ....comparison of real-life images, obtained using a slit lamp……

ANSWER: We have replaced “complex” with “relatively simple”. Also, we have edited the sentence as suggested. 

67-72: This is a very complex, composite sentence that is difficult to understand. Can it be simplified?

ANSWER: We have edited and simplified this sentence.

79-81: What is the difference between having an advanced knowledge of contact lens complications (high intensity) and having been additionally trained in ocular pathology (specificity)? Seems that these would be the same thing.

ANSWER: This refers to a previous research effort by some of the authors of the present study (Cardona G, Serés C. Grading contact lens complications: the effect of knowledge on grading accuracy. Curr Eye Res. 2009;34:1074-1081). The basis for that study originated in the syllabus implemented in our BSc in Optics and Optometry. In this regards, the course in ocular pathology provided students with knowledge on ocular health and disease, irrespective of whether or not the ocular condition was related to contact lens wear. On the other hand, during the three progressively advanced contact lens courses students were trained to diagnose, grade and manage complications caused by contact lens wear. A student having completed only the ocular pathology course may diagnose a certain ocular complication, but may not be able design an appropriate management strategy if the complication is associated with contact lens wear. Similarly, a student having completed only the contact lens courses will be able to offer a progressively better diagnose, grade and management of the contact lens related complication, but will lack the pathophysiological background to place the condition in the proper context. We believed that the effect of both aspects, although with a moderate amount of overlapping, could be studied independently. 

82: ….the Brien Holden……

ANSWER: Thank you. We have edited the text. 

92: “The aim of the present study……explore additional sources of bias….”. The Introduction is unsatisfactory because, while it discusses some aspects of using grading scales (intra- and inter- user variability, tendency to use increments etc.), it does little to explain how the outcomes might be subject to bias. Thus, the suggestion here that the aim of this study is “exploring additional sources of bias” is unconvincing. Variability and bias are different phenomena and in the context of this manuscript, the Introduction should be focused on the latter rather than the former.

ANSWER: We have edited the Introduction section of the manuscript extensively to better address the topic of the current research. Thank you for your comment. 

92: Is the term "awareness of disease progression" the most appropriate to describe what is being assessed here? What the graders are being told is whether that eye has been treated or not. They do not know anything about how the disease is progressing, just that someone has done something. Consequently the title of the paper may be misleading and should be changed.

ANSWER: The reviewer is correct. We have replaced “awareness of disease progression” with “awareness of treatment”, both in the tittle of the manuscript and elsewhere in the body of the text. 

102-109: Please verify that informed consent was obtained from participants.

ANSWER: We have added this information to the Subjects section of the revised manuscript: “Written informed consent was provided by all participants after the nature of the study was explained to them. In this regards, at the start of the study, subjects were only informed that they would be participating in three grading sessions. It was only following the conclusion of the third grading session that the full purpose of the study was explained, including the partial deceit required for the third grading session. “ 

Did you inform the participants that they should attempt to carry out the grading according to the scale provided and irrespective of the label on the images? This is an important factor as conscious bias may be more pronounced than unconscious bias.

ANSWER: Participants were not specifically directed to the need to grade images without paying attention to the label on the images, that is, the actual relevance of the label within the purpose of this study was not highlighted. Participants were informed that they should grade images as they had done in the previous two sessions, with the only difference that in the third session images corresponded to “treated” or “untreated” conditions. Thus, we believe that bias was unconscious. We have provided more information in the revised manuscript.

In general, the statistical analysis seems overly complicated to achieve the stated aims. Bland & Altman (Bland, J Martin, and Douglas G Altman. 1999. 'Measuring agreement in method comparison studies', Statistical Methods in Medical Research, 8: 135-60) have given clear guidelines on how to investigate these issues and as these methods are being used (though without appropriate citations, please add), it isn’t obvious why additional and more convoluted analyses are necessary. For any given condition (or all conditions combined) all that is needed to test the hypothesis that treatment knowledge affects grading scores is:

i. First, show that the first two repeats (R1 and R2) are unbiased, i.e. the mean difference (R1-R2), for all observers, is not significantly different from zero.

ii. Second, show that the third repeat (R3) is significantly biased, by demonstrating that the mean differences (R1-R3) and/or (R2-R3) are significantly different from zero.

Both i) and ii) above can be achieved by observing the 95% confidence interval for the mean differences and inspection of the scatter of points in the plots may add additional insight. It isn’t clear what extra value is offered by ANOVA, correlation, ICCs etc., so these procedures need to be clearly justified, if they are to be included.

ANSWER: We would like to express our gratitude to the reviewer for this comment. We acknowledge that the statistical analysis was too complicated and redundant. Therefore, we have simplified it by deleting one of the two calculations of grading reliability (see response below), as well as the ANOVA and post-hoc analysis of differences in grading reliability amongst conditions (certainly not relevant for the purpose of the study). We have opted not to remove the intraclass correlation analysis, as this is useful to determine whether participants offered reliable grades, when comparing session 1 and session 2. Non-reliable grades between these two sessions would probably not recommend further analysing the performance of these participants. In addition, we have provided the 95% confidence intervals for the mean differences between grading sessions. Finally, we have added the needed reference to the Bland and Altman paper. 

191: Why is the precision quoted for limbal vascularization different from the other features (integer vs one decimal place)?

ANSWER: We have corrected this error. All precision values are quoted with one decimal place in the revised manuscript. The correct value for limbal vascularization is 0.0. 

191-194: Can you explain what the difference is between “test-retest differences” and test-retest discrepancy”? These sound like the same thing, but as the SDs being quoted are different, presumably they are not? Is the ”average value of the test-retest differences” not the same as the “average value for the test-retest discrepancy distribution”? Some elaboration and clarification is required.

ANSWER: It is easier to explain it with an example. Average values for test-retest grading differences were 0.2 ± 1.8 (mean ± SD) for bulbar hyperaemia. On the other hand, SD and 95% confidence intervals of the test-retest discrepancy distributions were ±4.2 (±3.7 to ±4.7) for bulbar hyperaemia. As we had 8 images of bulbar hyperaemia and 30 participants, to determine the first set of values (0.2 ± 1.8), we calculated the mean (session 1-session 2) of the 8 images for each participant, and then we calculated the mean and SD of this value including all participants (n=30). This gave us the mean and SD of the differences between session 1 and session 2 for each condition and all participants. Similarly, for the second set of values [±4.2 (±3.7 to ±4.7)], we obtained the mean and 95% confidence intervals of the 30 values of SD obtained from the 8 differences between session 1 and session 2. According to (Efron N, Morgan PB, Jagpal R. The combined influence of knowledge, training and experience when grading contact lens complications. Ophthalmic Physiol Opt. 2003;23:79-85) the widespread of this second set of values gives an indication of grading reliability, with smaller values of SD denoting better reliability. 

As reliability was also explored with the intraclass correlation coefficient (ICC), we have opted to delete part of this information: it is a potentially confusing and redundant estimation of grading reliability. In addition, we have edited the text accordingly. 

246-267: This section appears to have nothing to do with the study data or outcomes. Please consider deleting it to shorten the text.

268-289: Likewise, this section repeats information already given elsewhere in the manuscript and can be deleted.

ANSWER: We have edited the Discussion of the manuscript to avoid redundancy and to remove irrelevant sections. 

344-351: Although this is presented as the Conclusion, most of the text does not rely on or “conclude” from data presented in the manuscript. For example, the text between lines 344 and 348 makes a series of unrelated statements about clinical practice. Please revise to ensure conclusions are relevant and meaningful.

ANSWER: We have edited the Conclusions section of the manuscript as follows: “In conclusion, many sources of bias have been reported to influence grading precision and reliability. The present findings revealed a statistically significant bias, referred to as “awareness of treatment”, in which examiners with moderately reliable grading skills tended to award higher grades to “untreated” conditions and lower grades to “treated” conditions. Further research is required to investigate the clinical significance of this source of bias, particularly given the confined characteristics of the sample and grading conditions of this study.” 

The use of the word “novel’ at lines 94 and 348 is unreasonable and should be deleted. Types of bias have always existed; it is only our awareness of them that is new.

ANSWER: We have removed the term “novel” from the manuscript. 

Check Ref 11: Spelling of Downie.

ANSWER: Thank you for pointing this typo to us. We have edited the text.

---

## [Decision Letter · Decision Letter 1]

15 Oct 2019

PONE-D-19-17494R1

Awareness of treatment: a source of bias in subjective grading of ocular complications

PLOS ONE

Dear Dr Cardona,

Thank you for submitting your manuscript to PLOS ONE. After careful consideration, we feel that it has merit but does not fully meet PLOS ONE’s publication criteria as it currently stands. Therefore, we invite you to submit a revised version of the manuscript that addresses the points raised during the review process.

The authors have addressed many of the issues raised in the earlier reviews, however there arel a few issues that remain to be resolved.

1. How were the Intraclass Correlation Coefficients calculated, and are these necessary given repeats one and two were analysed; please justify the use of this analysis. 

2. Please report all data with significant decimal places that relates or reflects the precision of the measurement being taken. For example, grading scales to 2 significant places does not reflect the precision of grading scales *per se.*

3. Please address the issue of the practical and clinical significance of the overall study findings (Reviewer 2).

We would appreciate receiving your revised manuscript by Nov 29 2019 11:59PM. To enhance the reproducibility of your results, we recommend that if applicable you deposit your laboratory protocols in protocols.io, where a protocol can be assigned its own identifier (DOI) such that it can be cited independently in the future. For instructions see: http://journals.plos.org/plosone/s/submission-guidelines#loc-laboratory-protocols

We look forward to receiving your revised manuscript.

Kind regards,

Michele Madigan

Academic Editor

PLOS ONE

Reviewers' comments:

Reviewer's Responses to Questions

**Comments to the Author**

1. If the authors have adequately addressed your comments raised in a previous round of review and you feel that this manuscript is now acceptable for publication, you may indicate that here to bypass the “Comments to the Author” section, enter your conflict of interest statement in the “Confidential to Editor” section, and submit your "Accept" recommendation.

Reviewer #1: All comments have been addressed

Reviewer #2: (No Response)

2. Is the manuscript technically sound, and do the data support the conclusions?

Reviewer #1: (No Response)

Reviewer #2: Partly

3. Has the statistical analysis been performed appropriately and rigorously? 

Reviewer #1: (No Response)

Reviewer #2: Yes

4. Have the authors made all data underlying the findings in their manuscript fully available?

Reviewer #1: (No Response)

Reviewer #2: No

5. Is the manuscript presented in an intelligible fashion and written in standard English?

Reviewer #1: (No Response)

Reviewer #2: Yes

6. Review Comments to the Author

Reviewer #1: (No Response)

Reviewer #2: I thank the authors for their responses and have the following further comments.

As this paper is about bias, I wonder whether ICC statistics add any useful information. I agree that establishing repeatability by comparing R1 and R2 is an important step, however aren’t the mean difference between R1 and R2 and the LOAs sufficient to do this? If the authors insist on quoting ICCs, they should specify that the correlations are being calculated between R1 and R2. (lines 173-185). Also presumably ICCs were calculated for each image? Please clarify.

There is still some inconsistency in precision. LOA values have 2 decimal places as quoted and ICC values have three. Note that the point here is not that they should all be the same, but that they reflect what is reasonable to expect from the characteristics of the measurement. How precise can a subjective grading instrument really be?

I am very confused by the response concerning the distinction between “test-retest differences” and “test-retest discrepancy”. You say “As we had 8 images of bulbar hyperaemia and 30 participants, to determine the first set of values (0.2 ± 1.8), we calculated the mean (session 1-session 2) of the 8 images for each participant, and then we calculated the mean and SD of this value including all participants (n=30). This gave us the mean and SD of the differences between session 1 and session 2 for each condition and all participants. Similarly, for the second set of values [±4.2 (±3.7 to ±4.7)], we obtained the mean and 95% confidence intervals of the 30 values of SD obtained from the 8 differences between session 1 and session 2.” But is it not true that, in general, the grand mean of a data set is invariant to the order in which the columns and rows are averaged? In other words, it shouldn't matter if you average first across the 8 images and then over the 30 participants, or the other way around, the mean (though not the SD) will be the same. Why then, are there two different mean values quoted here?

In choosing Bland Altman plots to show the data it might help readers see the differences being reported if the same Y-axis scale were used for the treated vs untreated data. Actually, a simple bar chart might be even easier to understand (one bar each for R1, R2 and R3 treated R3 untreated).

One final comment is that the quoted difference between R1 and R3 (lines 193-197) all seem to be well within the LOAs for R1 vs R2, irrespective of “treatment” bias. This could reasonably be interpreted to suggest that the typical amount of bias introduce by treatment knowledge is smaller than the normal range of variation found when making successive judgments of the same image. What does this mean for the practical significance of the findings as well as the conclusion that “…more research is needed to investigate the clinical significance.”? Is there not sufficient evidence here to conclude that the effect is unlikely to be of clinical significance?

7. PLOS authors have the option to publish the peer review history of their article (what does this mean?). If published, this will include your full peer review and any attached files.

Reviewer #1: No

Reviewer #2: No

---

## [Author Response · Author response to Decision Letter 1]

27 Nov 2019

Manuscript No.: PONE-D-19-17494R1

Title: Awareness of treatment: a source of bias in subjective grading of ocular complications

PLOS ONE

All changes are marked in red in the manuscript. We would like to thank the reviewer for their constructive and positive comments to our manuscript.

Academic Editor

1. How were the Intraclass Correlation Coefficients calculated, and are these necessary given repeats one and two were analysed; please justify the use of this analysis. 

2. Please report all data with significant decimal places that relates or reflects the precision of the measurement being taken. For example, grading scales to 2 significant places does not reflect the precision of grading scales per se.

3. Please address the issue of the practical and clinical significance of the overall study findings (Reviewer 2).

ANSWER: Dear Academic Editor, we have addressed these concerns according to the comments and suggestions of the reviewer (please refer to the responses provided below). 

Reviewer #2

I thank the authors for their responses and have the following further comments.

As this paper is about bias, I wonder whether ICC statistics add any useful information. I agree that establishing repeatability by comparing R1 and R2 is an important step, however aren’t the mean difference between R1 and R2 and the LOAs sufficient to do this? If the authors insist on quoting ICCs, they should specify that the correlations are being calculated between R1 and R2. (lines 173-185). Also presumably ICCs were calculated for each image? Please clarify.

ANSWER: We have edited the Data Analysis section of the manuscript to better describe how ICC values were calculated: “Intraclass correlation coefficients (ICC) (two-way mixed effects, absolute agreement, multiple examiners/measurements model) of session 1 and 2 (test-retest) were calculated to determine grading reliability for each image and the corresponding median and range (minimum-maximum) of values for each ocular condition (comprised of 8 images) was determined”. Even if redundant, we have also added this information to the Results section.

We believe that ICC values are relevant, as they provide an estimation of reliability with a single numerical value that can be compared with published normative data. The Bland-Altman analysis, on the other hand, is very useful in showing the actual distribution of the differences between two measurements, and highlighting any possible bias. We agree with the reviewer that both analyses provide information about reliability, but we believe that this information is not redundant, but complementary. If the reviewer considers that it is indeed unnecessary, however, we would not oppose deleting the ICC analysis from the manuscript.

There is still some inconsistency in precision. LOA values have 2 decimal places as quoted and ICC values have three. Note that the point here is not that they should all be the same, but that they reflect what is reasonable to expect from the characteristics of the measurement. How precise can a subjective grading instrument really be?

ANSWER: The reviewer is correct. We have attempted to be consistent with the number of significant digits throughout the manuscript. Thus, we have followed the recommendations of [Koo TK, Li MY. A guideline of selecting and reporting intraclass correlation coefficients for reliability research. J Chiropr Med. 2016;15:155-163] and [Liljequist D, Elfving B, Roaldsen KS. Intraclass correlation – A discussion and demonstration of basic features. PLoS One. 2019;14:e0219854], as well as of [https://labwrite.ncsu.edu/res/gh/gh-sigdig.html], and reported ICC values to two significant digits (such as 0.80 or 0.75) and subjective grading scores to one significant digit, that is, the tenths digit (such as 23.0 or 19.2). We have added the reference [Liljequist D et al] to our manuscript.

I am very confused by the response concerning the distinction between “test-retest differences” and “test-retest discrepancy”. You say “As we had 8 images of bulbar hyperaemia and 30 participants, to determine the first set of values (0.2 ± 1.8), we calculated the mean (session 1-session 2) of the 8 images for each participant, and then we calculated the mean and SD of this value including all participants (n=30). This gave us the mean and SD of the differences between session 1 and session 2 for each condition and all participants. Similarly, for the second set of values [±4.2 (±3.7 to ±4.7)], we obtained the mean and 95% confidence intervals of the 30 values of SD obtained from the 8 differences between session 1 and session 2.” But is it not true that, in general, the grand mean of a data set is invariant to the order in which the columns and rows are averaged? In other words, it should not matter if you average first across the 8 images and then over the 30 participants, or the other way around, the mean (though not the SD) will be the same. Why then, are there two different mean values quoted here?

ANSWER: As suggested by the reviewer and noted in the previous responses to the reviewers “As reliability was also explored with the intraclass correlation coefficient (ICC), we have opted to delete part of this information: it is a potentially confusing and redundant estimation of grading reliability”, this information was considered redundant and deleted from the revised manuscript (R1). 

Aiming at clarifying this issue, we agree with the reviewer that the grand mean of a data set is invariant to the order in which columns and rows are averaged. However, please note that the set of values [±4.2 (±3.7 to ±4.7)] did not correspond to the grand mean of grading scores (that is, 8 images per 30 participants) but to the mean (of the 30 participants) of the SD obtained from the differences between R1 and R2 of each condition (set of 8 images) and participant. 

In choosing Bland Altman plots to show the data it might help readers see the differences being reported if the same Y-axis scale were used for the treated vs untreated data. Actually, a simple bar chart might be even easier to understand (one bar each for R1, R2 and R3 treated R3 untreated).

ANSWER: Thank you. For clarity, we have added a bar chart figure displaying all results in the same scale (Figure 3).

One final comment is that the quoted difference between R1 and R3 (lines 193-197) all seem to be well within the LOAs for R1 vs R2, irrespective of “treatment” bias. This could reasonably be interpreted to suggest that the typical amount of bias introduce by treatment knowledge is smaller than the normal range of variation found when making successive judgments of the same image. What does this mean for the practical significance of the findings as well as the conclusion that “…more research is needed to investigate the clinical significance.”? Is there not sufficient evidence here to conclude that the effect is unlikely to be of clinical significance?

 ANSWER: The reviewer highlights an important issue. We have further addressed the clinical significance of our findings in the Discussion section of the manuscript (and edited the conclusions of the abstract). The conclusion of the revised manuscript is as follows: “In conclusion, many sources of bias have been reported to influence grading precision and reliability. The present findings revealed a statistically significant bias, referred to as “awareness of treatment”, in which examiners with moderately reliable grading skills tended to award higher grades to “untreated” conditions and lower grades to “treated” conditions. As the study was designed, with very homogeneous characteristics in sample and grading conditions, the clinical significance of this source of bias could not be considered as manifestly superior to the normal range of variation found when making successive judgments of the same image.”

---

## [Editor Report · Decision Letter 2]

11 Dec 2019

Awareness of treatment: a source of bias in subjective grading of ocular complications

PONE-D-19-17494R2

Dear Dr. Cardona,

We are pleased to inform you that your manuscript has been judged scientifically suitable for publication and will be formally accepted for publication once it complies with all outstanding technical requirements.

With kind regards,

Michele Madigan

Academic Editor

PLOS ONE
---

## [Editor Report · Acceptance letter]

16 Dec 2019

PONE-D-19-17494R2 

Awareness of treatment: a source of bias in subjective grading of ocular complications 

Dear Dr. Cardona:

I am pleased to inform you that your manuscript has been deemed suitable for publication in PLOS ONE. Congratulations! Your manuscript is now with our production department. 

With kind regards,

on behalf of

Dr. Michele Madigan 

Academic Editor

PLOS ONE